# Recent Advances in Drug Delivery System Fabricated by Microfluidics for Disease Therapy

**DOI:** 10.3390/bioengineering9110625

**Published:** 2022-10-29

**Authors:** Fuhao Jia, Yanbing Gao, Hai Wang

**Affiliations:** 1CAS Key Laboratory for Biomedical Effects of Nanomaterials & Nanosafety, CAS Center for Excellence in Nanoscience, National Center for Nanoscience and Technology, Beijing 100190, China; 2School of Nanoscience and Technology, University of Chinese Academy of Sciences, Beijing 100049, China; 3Troop 96901 of the Chinese People’s Liberation Army, Beijing 100094, China

**Keywords:** microfluidics, drug delivery, micro/nanoparticles

## Abstract

Traditional drug therapy faces challenges such as drug distribution throughout the body, rapid degradation and excretion, and extensive adverse reactions. In contrast, micro/nanoparticles can controllably deliver drugs to target sites to improve drug efficacy. Unlike traditional large-scale synthetic systems, microfluidics allows manipulation of fluids at the microscale and shows great potential in drug delivery and precision medicine. Well-designed microfluidic devices have been used to fabricate multifunctional drug carriers using stimuli-responsive materials. In this review, we first introduce the selection of materials and processing techniques for microfluidic devices. Then, various well-designed microfluidic chips are shown for the fabrication of multifunctional micro/nanoparticles as drug delivery vehicles. Finally, we describe the interaction of drugs with lymphatic vessels that are neglected in organs-on-chips. Overall, the accelerated development of microfluidics holds great potential for the clinical translation of micro/nanoparticle drug delivery systems for disease treatment.

## 1. Introduction

Microfluidic technology is a highly interdisciplinary science and technology, involving engineering, physics, chemistry, microfabrication, and other disciplines [1,2,3]. Microfluidics can process or manipulate tiny amounts (10^−9^ L to 10^−8^ L) of fluid within microchannels, ranging in size from tens to hundreds of micrometers. Since the heat and mass transfer processes at the microscale are scale-dependent, the fluids in microfluidic chips exhibit specific properties that are different from those at the macroscale [4]. In short, surface tension and capillary forces dominate the device, and the effects of gravity and inertial forces are negligible. The interface factors such as diffusion, surface tension and viscosity become the main factors affecting the fluid behavior [3]. In addition, the high surface area to volume ratio ensures thermal homogeneity and rapid heat transfer. These fundamental properties give rise to a wide range of advantages, including minimal use of reagents, low energy consumption, fast reaction rates, and massively parallel processes [5]. Thus, microfluidic devices are widely used in industrial and academic research.

Currently, both small drugs and macromolecular drugs require the appropriate drug delivery strategies [6]. For example, traditional small molecule drugs tend to have low solubility, which limits their bioavailability. Furthermore, the in vivo stability of protein drugs after administration is compromised by proteases, temperature and pH. Nucleic acid drugs need to be delivered into the cytoplasm to be effective. To address these challenges, it is necessary to develop appropriate drug delivery systems. Proper drug delivery allows not only on-demand delivery of active drug to target tissues or cells, but also proper control of pharmacokinetics (e.g., in vivo distribution, half-life, maximum concentration in serum, etc.) [7]. Although pharmacokinetics can be modulated by controlling the number and frequency of dosing, many disease treatments require frequent long-term dosing, which compromises patients’ quality of life. A variety of controlled release systems, including injectable hydrogels, polymer implants and micro/nanoparticles, have been used to improve drug delivery behaviors [8]. Drugs within a delivery system have multiple release modes, including drug diffusion, degradation of the delivery material, and external stimuli. Macroscopic delivery systems such as drug-eluting stents require surgical implantation and may induce fibrocystic reaction [9]. In addition, the macroscopic delivery system has a singular drug release pattern. Compared with macroscopic delivery systems, micro/nanoparticle delivery systems offer many advantages in protecting protein and nucleic acid drugs from degradation, controlling drug release profiles, enhancing small molecule solubility of hydrophobic drugs, and tunable tissue targeting [10].

Preparation of engineered micro/nanoparticles with controlled size, monodispersity, multiple morphologies, and specific functions plays an increasingly important role in biomedicine [11,12]. However, traditional micro/nanoparticle synthesis methods such as emulsion polymerization, dispersion polymerization, spray drying, etc., often produce micro/nanoparticles with high polydispersity, poor reproducibility, limited functions, and poorly regulated morphology [13]. Microfluidic technology provides a technique to precisely control and manipulate fluids to generate highly monodisperse droplet emulsions in a bottom-up manner [14,15]. The size, structure, function, and dispersion of micro/nanoparticle can be precisely controlled by the structure design of the microfluidic device, flow rate regulation, material composition modulation, and fluid viscosity selection [16,17]. Therefore, microfluidics provides a platform for the preparation of micro/nanoparticles and expand their application in drug delivery.

The main purpose of this review is to summarize the research on microfluidic preparation of drug delivery systems and lymphatics-on-a-chip. Specifically, this review focuses on the preparation of various stimuli-responsive micro/nanoparticles, including micro/nanogels, microcapsules, lipid nanoparticles, and hierarchical micro/nanoparticles. The construction of the lymphatic chip and its interaction with drugs are also introduced.

## 2. Microfluidic Device

We first introduce the material and processing technology of microfluidic devices before describing their applications. The foremost step for microfluidic applications is the selection of the appropriate materials. Depending on the application, factors to consider in material selection include durability, clarity, biocompatibility, chemical compatibility with application reagents, temperature, pressure, and surface modification [18]. A variety of materials have been developed to fabricate microfluidic chips, including silicon, glass, polymer, and paper [19,20]. Silicon is used to fabricate microfluidic devices due to its thermal stability, chemical stability, solvent resistance, and high thermal conductivity (Figure 1a) [20,21]. However, the opacity of silicon limits its application in optical detection. Furthermore, silicon material is very fragile, making it difficult to introduce valves or pumps on the device. An alternative material could be glass due to its chemical inertness, thermal resistance, electrical insulation, biocompatibility and easy functionalization of surfaces [20,22]. Glass capillaries can be assembled to form microfluidic devices to fabricate micro/nanoparticles (Figure 1b) [23]. A typical setup consists of three coaxially assembled basic modules: an injection tube, a transition tube, and a collection tube. Although the cost of glass is low, the process of fabricating chips from glass is time-consuming and labor-intensive [24,25,26]. In addition, the extremely impermeable properties of glass and silicon limit on-chip cell culture [27].

Alternatively, polymeric materials have been extensively explored to fabricate microfluidic devices. Compared to silicon and glass, polymeric materials have the advantages of lower cost, simple processing technology, high transparency and high thermal resistance [28]. A representative material is polydimethylsiloxane (PDMS), an optically transparent soft elastomer (Figure 1c) [29]. PDMS is relatively inexpensive, easy to mold, and possesses properties such as light transmission, air permeability, biocompatibility, natural hydrophobicity, and high elasticity [30,31]. Therefore, it is suitable for long-term cell culture, cell screening and biochemical assays. At the same time, high flow rates lead to microchannel expansion [32]. PDMS may adsorb organic solvents that cause deformation of the microchannels. In addition to PDMS, polymethyl methacrylate, polyvinyl chloride, fluoropolymers, and cyclic olefin polymers have also been employed for microfluidic device fabrication [33,34,35,36].

Paper-based microfluidic devices represent a new type of microfluidic system developed in recent years, with the advantages of low cost, reproducibility, easy fabrication and handling, portability, and easy integration with other devices [37]. The flow of fluids in paper is primarily controlled by adhesion and cohesion forces that create capillary action in the cellulose matrix. Therefore, the fluid can be precisely directed by hydrophobic modification of certain regions in the matrix (Figure 1d) [38]. The inherent micro/nanostructure of the paper matrix can be used to simulate the cellular microenvironment, such as oxygen/nutrient gradients, shear forces, etc. [39]. Thus, paper can serve as an attractive 3D scaffold to culture cells.

As mentioned above, there are a variety of materials used to fabricate microfluidic devices. It is also essential to select the appropriate processing method according to the material properties and product requirements. In this review, we highlight several widely used fabrication techniques. Wet and dry etching techniques are commonly used in silicon and glass processing technologies (Figure 1e) [40]. Wet etching is well-regarded for its fast etch speed. However, hydrofluoric acid is mostly used as an etchant, which is highly corrosive and harmful to the environment. In addition, wet etched channels tend to exhibit an isotropic distribution. In contrast, dry etching (also known as reactive ion beam etching) utilizes an ion beam to bombard a substrate, creating anisotropic, dimensionally accurate channels. However, dry etching is slow and expensive to process. Micromachining, including mechanical cutting, abrasive jet processing, and the ultrasonic processing method, has also been used to process silicon and glass, but the precision and productivity of micromachining are relatively modest.

Soft lithography (also known as replica molding) is also one of the most commonly used microfluidic chip fabrication techniques [41]. Traditional fabrication processes for PDMS microfluidics devices include photomask design, photoresist spin coating, optical exposure, development and formation of PDMS replicas, and sealing (Figure 1f) [42]. In brief, to obtain the desired device, the first process is to design a patterned photomask. The photoresist is then spin coated onto the silicon wafer substrate. Then, the coated silicon wafer is heated to remove the solvent from the photoresist. A photomask is placed on top of the photoresist and both are exposed to UV light. After removing non-crosslinked areas, PDMS is poured to form the rubbery layer. Then, two peeled PDMS or PDMS and glass are sealed after oxygen plasma treatment. Finally, the device is connected to the pipeline for liquid flow. Soft lithography technology has high processing accuracy and small differences in the geometry of different devices, which can improve the repeatability of experiments and increase the credibility of results [43]. The limitations of soft lithography include pattern distortion during method replication, and the process requires a dedicated clean room.

Due to its unique properties, 3D printing has emerged as one of the alternative manufacturing methods for fabricating microfluidic devices (Figure 1g) [44]. There are several conventional 3D printing techniques, including inkjet 3D printing, fused deposition modeling, stereolithography, and two-photon polymerization techniques [45]. Three-dimensional printing can fabricate microfluidic chips with complex three-dimensional geometries and permits the rapid adjustment of the functions of the device by optimizing the design. In addition, it allows high throughput fabrication of microfluidic devices [46]. Importantly, as technology advances, the accuracy of 3D printing continues to improve and is approaching the accuracy of soft lithography. The majority of 3D printing materials are based on acrylates and acrylonitrile butadiene styrene, which can lead to undesired biotoxicity [47,48]. The compatibility of 3D printing materials with solvents remains a challenge. For example, some materials undergo swelling in aqueous environments.

In summary, each material and processing technology has its own challenges and limitations. It is critical to select the appropriate materials and fabrication techniques to fabricate microfluidic devices based on the desired properties and potential application.

**Figure 1 bioengineering-09-00625-f001:**
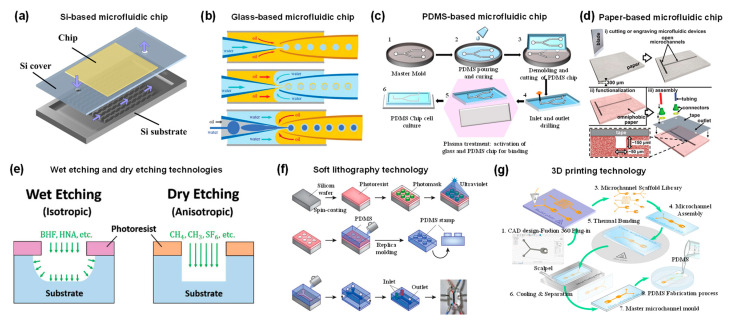
(**a**) Schematic diagram of silicon-based microfluidic device. Reproduced with permission [23]. Copyright 2015, Multidisciplinary Digital Publishing Institute. (**b**) Schematic illustration of the coaxial capillary microfluidic devices. Reproduced with permission [23]. Copyright 2018, Royal Society of Chemistry. (**c**) Fabrication process of PDMS devices by replica molding method. Reproduced with permission [29]. Copyright 2020, Multidisciplinary Digital Publishing Institute. (**d**) Schematic diagram of paper-based microfluidic device fabrication. Reproduced with permission [38]. Copyright 2013, Royal Society of Chemistry. (**e**) Schematic diagram of anisotropic wet etching and anisotropic dry etching techniques for processing glass. Reproduced with permission [40]. Copyright 2018, Wiley. (**f**) Schematic illustration of a typical PDMS microfluidic device fabricated by soft lithography. Reproduced with permission [42]. Copyright 2014, Springer Nature. (**g**) Schematic illustration of the 3D printing technique to fabricate microfluidic molds. Reproduced with permission [44]. Copyright 2021, Public Library of Science.

## 3. Drug Delivery System Fabricated by Microfluidics

Compared with a traditional drug delivery system, micro/nanoparticles can improve stability, solubility, and circulation time in vivo [49]. In order to improve the therapeutic efficacy and reduce the side effects of drugs, different kinds of micro/nanocarriers have been developed using biocompatible materials, including microgels, microcapsules, nanoliposomes, nanomicelles, and nanoemulsions [50,51]. However, conventional synthetic methods fail to obtain micro/nanocarriers with uniform size, shape and composition. Microfluidic devices offer an alternative option to address this problem. In general, the fabrication of microparticles is mainly based on the manipulation of immiscible fluids on droplet microfluidics to generate monodisperse droplets. The fabrication of nanoparticles is mainly based on the rapid mixing of multiphase fluids in microfluidics mixer. This section summarizes the preparation of micro/nanoparticles based on droplet microfluidics, centrifugal microfluidics and microfluidic mixers.

### 3.1. Micro/Nanogels for Drug Delivery

Hydrogels have high water content, tunable chemical and physical structure, excellent mechanical properties, and biocompatibility [52]. Micro/nanogels are three-dimensional crosslinked polymer particles with the features of hydrogels and colloidal particles [53]. Typical materials used to prepare hydrogels include natural polymers (such as alginates, chitosan, hyaluronic acid, etc.) and synthetic polymers (such as polyvinyl alcohol, polyethylene glycol, polyacrylamide, polyhydroxyethylmethacrylate, etc.). The crosslinking methods of hydrogels include ionic crosslinking, covalent crosslinking, UV curing, photoinitiation, electrostatic complexation, interfacial assembly, etc. [54]. Due to the advantages of tunable size, large surface area, abundant internal macromolecular network, and good biocompatibility, micro/nanogels can be candidates for clinical therapeutic drug delivery [55]. In addition, micro/nanogels can effectively protect the encapsulated drugs from the external environment and transport the drugs to specific tissues and/or cells through surface modification. 

Alginate hydrogels have been widely used as drug carriers due to their excellent biocompatibility. It is well known that a method available for alginate hydrogels is to crosslink alginate with divalent cations such as Ca^2+^ and Ba^2+^ [56]. For example, Cai et al. reported a concentration-controlled microfluidic chip to synthesize DOX-encapsulated alginate nanogels (CaA@Dox) [57]. As shown in Figure 2a, alginate and Ca^2+^ ions diffused into the central channel driven by a concentration gradient, forming alginate nanogels. The diameter of alginate microgels is positively related to the ratio of channel lengths and flow rate. With the increase of pH value, the carboxyl group is continuously dissociated, the hydrophilicity of sodium alginate increases, and the molecular chain is elongated. Therefore, sodium alginate has significant pH sensitivity. Sodium alginate microgels can respond to the acidic microenvironment of tumors.

In addition to microfluidic chips based on a diffusion mixing mode, centrifugal microfluidic chips are also applied to generate high-throughput centrifugal droplets [58,59]. Kim et al. utilized photo-crosslinkable methacrylate-modified hyaluronic acid microgels using a portable centrifugal microfluidic chip [60]. In the initial state, aqueous solution and oil were injected into the inlet and outlet chambers of the microchip, respectively. Due to the different specific gravity and immiscibility of oil and water, droplets were generated at the oil–water interface under the action of centrifugal force (Figure 2b). Parallel microchannels at the oil–water interface were designed to restrict the droplet size.

By exploiting the environmental sensitivity of polymer networks, hydrogels can be easily designed as stimuli-responsive carriers [61,62]. Depending on the source, stimuli can be divided into physiological environmental stimuli (i.e., tumor acidic microenvironment, inflammation or enzymes at the lesion site) and external stimuli (i.e., temperature, light or magnetic field) [63]. Poly(N-isopropylacrylamide) (PNIPAM) is a representative thermosensitive polymer [64]. PNIPAM shrinks and changes from hydrophilic state to hydrophobic state when the temperature is higher than its lower critical solution temperature (LCST) [65]. Luo et al. fabricated photosensitive microgels using PNIPAM polymers and polypyrrole (PPy) nanoparticles via a co-flow microfluidic chip (Figure 2c) [66]. PPy nanoparticles have excellent photothermal conversion properties. After the first irradiation cycle, PNIPAM microgels containing PPy nanoparticles released 45% of FITC-albumin, which was significantly higher than the control group. Among the various stimulus responses, enzyme-triggered drug release is an attractive approach [63]. Enzymatic reactions provide higher selectivity and specificity due to the specificity of the enzymes. Enzymes play an integral role in physiological processes and are involved in all biological processes. Malignant diseases or lesions lead to elevated expression of specific enzymes in certain tissues. Several natural polymers such as fibrin, collagen, gelatin, and hyaluronic acid have been used to prepare enzyme-responsive hydrogels. Hyaluronidase is overexpressed in aggressive malignancies or secreted by pathogenic bacteria at the site of infection. Busatto et al. synthesized an oil-in-water nanoemulsion to encapsulate hydrophobic drug progesterone [67]. The nanoemulsion and hyaluronic acid precursor solution were mixed in a microfluidic chip to form an oil-in-water-in-oil nanostructure (Figure 2d). Ultimately, hyaluronic acid precursors were crosslinked under ultraviolet light and formed microgels. When exposed to a concentration of 100 UI/mL of hyaluronidase, the hyaluronic acid microgels sustained drug release over two days. Another typical degradative enzyme is matrix metalloproteinase. Previous study showed that PEG microgels were prepared by using a matrix protease-degradable crosslinked peptide and dithiothreitol as the crosslinking agent [68]. The degradation rate of microgels in matrix metalloproteinase can be regulated by changing the ratio of crosslinking peptide to a dithiothreitol crosslinking agent.

**Figure 2 bioengineering-09-00625-f002:**
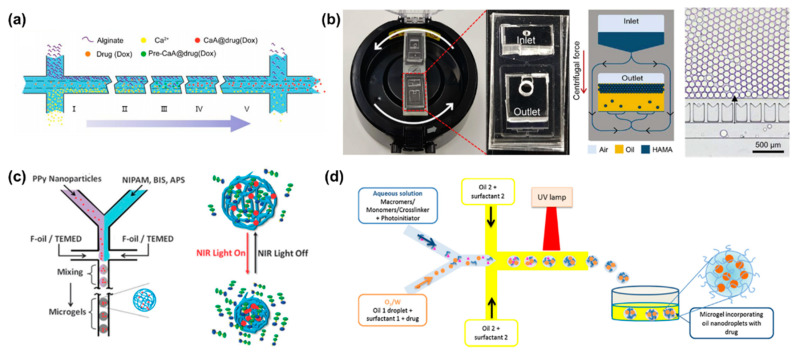
(**a**) Schematic illustration of microfluidic diffusion mixing mode, where alginate and Ca^2+^ ions diffuse into the intermediate channel and mix to form nanoparticles. Reproduced with permission [57]. Copyright 2019, Switzerland. (**b**) Photograph of a portable centrifugal microfluidic system assembled on a commercial centrifuge. Schematic diagram of microdroplet formation during centrifugation, and optical micrographs of microdroplets prepared in the microfluidic device. Reproduced with permission [60]. Copyright 2020, Elsevier. (**c**) Schematic illustration of PPy nanoparticles mixed with PNIPAM and served as photosensitive drug carriers fabricated with droplet microfluidics. Reproduced with permission [66]. Copyright 2013, Royal Society of Chemistry. (**d**) Schematic illustration of the preparation of biodegradable HA microgels with O/W/O as a template. Reproduced with permission [67]. Copyright 2017, Elsevier.

Overall, the micro/nanogels are able to drastically change their volume to release their internal drug in response to environmental changes. The chemical composition, size, biodegradability, morphology and surface functional properties of the micro/nanogels can be adjusted by the synthesis conditions such as monomer selection, monomer ratio, crosslinker concentration and initiator concentration. However, micro/nanogels have a homogeneous architecture, making it difficult to achieve elaborate functions. The clinical application of micro/nanogels remains challenging due to the limitations that exist during conventional batch synthesis, such as limited scalability, little control over reaction parameters, wide nanoparticle polydispersity, poor batch-to-batch reproducibility, and the need to use large amounts of chemicals and therapeutic agents. Microfluidic devices can generate discrete volumes of droplets to from micro/nanogels. Each droplet works as a microreactor, ensuring fast heat and mass transfer within a limited microvolume and enabling faster reaction kinetics. Thus, microfluidic devices precisely control the reaction parameters, such as the size and composition of droplets, to overcome constraints of conventional methods.

### 3.2. Microcapsules for Drug Delivery

In contrast to micro/nanogels, microcapsules have a core-shell structure where the core can be solid, liquid or gas [69,70]. Microcapsules do not exhibit sudden volume changes in the same way as micro/nanogels, due to shell support [71]. The high versatility of the shell materials allows the production of microcapsules with versatile functionality, such as enhanced retention, controlled release, and stimulus responsiveness [72,73]. The well-designed microcapsule shells not only protect encapsulated drugs, but also rupture and release the internal drugs in response to external stimuli, including pH, temperature, osmotic pressure, electric fields and stress [74]. 

Stimuli-responsive microcapsules have been widely applied in drug delivery systems. Kim et al. prepared microcapsules based on water/oil/water (W/O/W) double emulsion droplets, showing molecular polarity and temperature-dependent permeability [75]. The intermediate oil phase contains the photocurable monomer and the phase change material dodecanol (Figure 3a). Under UV light, dodecanol remained liquid and occupied the microcapsule space without polymerization. Very different from physical voids, the permeability of microcapsules depends on molecular size and molecular polarity [76]. In this case, low polarity molecules can be dissolved in water and transferred to the other side of the shell by transferring within dodecanol. However, the transport of highly polar molecules is inhibited. Similar to other phase-change materials, dodecanol melts at temperature above its melting point. Therefore, dodecanol freezes reversibly when the temperature is below the melting point. The release of ICG from microcapsules was minimal at 4 °C for 72 h, while the in vitro release process accelerated rapidly at 37 °C. The smart design of the shell material endows the microcapsules with polarity-dependent and temperature-dependent drug release patterns, providing a smart drug release strategy.

Electric fields can also be used to control the release of active substances in microcapsules by methods including electroporation and iontophoresis [77,78]. Depending on the location of the device, drug release near the skin can be triggered by a voltage generated by an external conductive skin patch. Alternatively, a minimally invasive implantable device controlled by radio frequency can be used to control drug release [79]. The microcapsules prepared by electro-responsive materials rupture was attributed to interface deformation caused by maximum well interface stresses at the droplet/dielectric interface induced by the applied alternating electric field (Figure 3b). During the interfacial deformation process, the fragile parts of the microcapsules were prone to defects, leading to rupture and eventually the release of internal drugs. Due to the competition between separation pressure and electrocompression, there is a critical voltage that causes microcapsules to rupture. The threshold voltages increased with the applied field frequency. In addition, high conductivity facilitated drug release at low voltage and was not limited by ion types. This capability can be extended to dual core droplets for drug co-delivery.

Preparation of temperature-responsive or pH-responsive microcapsules usually requires the addition of functional materials to the precursor solution, such as temperature-sensitive or pH-sensitive materials. However, osmolarity-triggered drug release avoids the addition of additional precursors [80]. Zhang synthesized osmotic pressure-responsive microcapsules with non-uniform shell thickness, which can be controlled by changing the flow rate ratio of the mesophase to the inner phase [81]. The thickness of the microcapsule shell varies widely, from 40 μm in the thickest part to 600 nm in the thinnest part (Figure 3c). As a result, non-uniform microcapsules are more prone to rupture from the thinnest areas than uniform microcapsules. It is also fascinating to impart multiple stimulation responses to single microcapsules. Multifunctional responsive microcapsules with a n-hexane layer were prepared with capillary microfluidic [76]. By inducing oil layer instability, hydrophilic drugs encapsulated in microcapsules were released on demand (Figure 3d). Ethanol solution, mechanical stress and osmotic pressure can all cause the oil layer rupture. However, the microencapsulated shell layer remained intact. 

**Figure 3 bioengineering-09-00625-f003:**
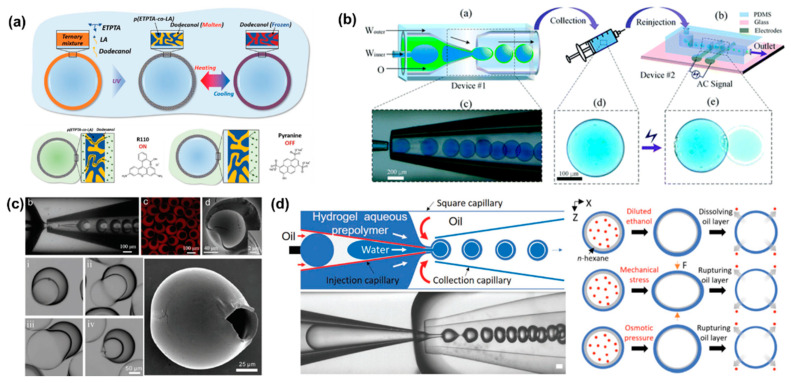
(**a**) Schematic illustration of microcapsule formation by double emulsion droplet templates and reversible phase change of dodecanol in shell voids during heating and cooling. Reproduced with permission [75]. Copyright 2019, Wiley. (**b**) Schematic illustration of co-flow capillary microfluidics chip for encapsulation and controlled release of drugs. The applied electric field can control the release of the internal drug from the microcapsule. Reproduced with permission [77]. Copyright 2018, Royal Society of Chemistry. (**c**) The optical images of capillary microfluidic chip preparation of inhomogeneous microcapsules. Confocal images, SEM images and optical images of inhomogeneous microcapsules. The optical images showed that the thinnest part of the microcapsule shell starts to swell and the volume increased until it ruptures. Reproduced with permission [81]. Copyright 2019, Wiley. (**d**) Schematic and optical image of microfluidic chip to produce microencapsulated cargos with a thin oil layer, where different external stimuli including dissolution, mechanical pressure and osmotic pressure induce destabilization of the interstitial oil layer to release the cargo. Reproduced with permission [76]. Copyright 2021, Wiley.

For malignant diseases, multidrug codelivery strategies tend to have a significant efficacy. Encouraging progress has been made in the design of stimuli-responsive microcapsule shells based on intrinsic properties of materials. Programmable drug release can be combined with multiple stimulation sequences to achieve a combination of different release modalities, such as burst release and sustained release. Yang et al. designed a multi-stimulus-responsive microcapsule structure consisting of a pH-responsive chitosan hydrogel shell and an oil core containing curcumin and drug-loaded poly(lactic acid-ethanolic acid copolymer) (PLGA) nanoparticles (Figure 4a) [82]. In pH solution of 1.5, the microcapsules ruptured rapidly within 60 s, which corresponds to the gastric environment. Since PLGA is semipermeable, the drug-loaded PLGA nanoparticles achieved sustained release in a neutral environment, with only 63.6% of curcumin released from the nanoparticles in 28 days. The combination of burst release and sustained release provides an effective treatment for acute diseases, with burst release alleviating the acute symptoms and sustained release maintaining the therapeutic effects. More complex Trojan-like microcapsules were prepared in a O1/W2/O3/W4/O5 quadruple microemulsion as templates for complex programmed release [83]. Functional shell materials were incorporated into the W2 and W4 phases to create capsule-in-capsule structures (Figure 4b). Chitosan, poly (ethylene glycol) diacrylate (PEGDA), and PNIPAM constituted three Trojan-like stimulus-responsive microcapsules, including CS@CS microcapsules, PEGDA@CS microcapsules, CS@PNIPAM microcapsules. The pH and temperature can stimulate the chitosan shell and the PNIPAM shell, respectively, resulting in shell rupture. Meanwhile, PEGDA has no temperature or pH response, so the encapsulated drug can be released sustainably. In general, microcapsules containing two stimuli-responsive hydrogel shells are able to provide a flexible trigger mechanism for multifunctional programmed sequential release behavior.

Furthermore, monodisperse multiple emulsions can contain several smaller droplets inside [84,85]. A two-stage co-flow microfluidic device was constructed by a sequential and coaxial nested assembly of multiple glass capillaries (Figure 4c). The uniform droplets generated in the first capillary were encapsulated into larger droplets by the second capillary, forming the drop-in-drop structure [85]. The number, proportion and size of the innermost droplets can be precisely controlled by adjusting the capillary size and fluid flow rate. Higher-order multiple emulsions, such as triple, quadruple, and quintuple emulsions, can be further controllably produced by more complex nested structures.

The conventional preparation method of microcapsules is usually to make bulk emulsion first, and then form the shell of the capsule through interfacial polymerization, precipitation or coagulation. With the bulk emulsification method, it is difficult to control the size, shape and structure of the microcapsules, which greatly limits their potential clinical application. In contrast, microfluidic devices provide a new route for the construction and synthesis of microcapsules by generating emulsions. The structure of microcapsule can be precisely controlled by multiple emulsions through T-junction, flow-focusing, or co-flowing microfluidic devices. In addition, by using a microfluidic device, the drug encapsulation efficiency of the microcapsules can be greatly improved. 

**Figure 4 bioengineering-09-00625-f004:**
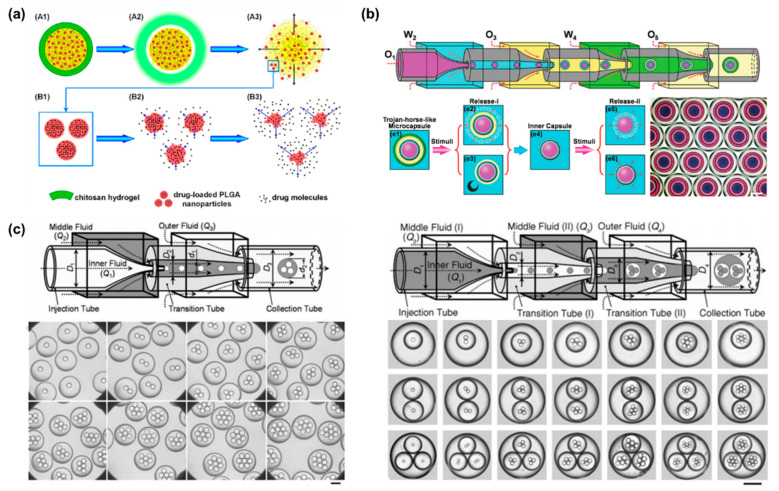
(**a**) Schematic illustration of programmed sequential drug release from multi-stimulus-responsive microcapsules. Acid triggered the decomposition of chitosan resulting in release of free drug and drug-loaded PLGA nanoparticles. Then PLGA degradation achieved sustained release. Reproduced with permission [82]. Copyright 2016, American Chemical Society. (**b**) Schematic illustration of a microfluidic chip prepared by sequential emulsification of Trojan-horse-like microcapsules with a capsular-in-capsule structure. Two-stage sequential release of microcapsules were triggered by different stimuli. Reproduced with permission [83]. Copyright 2018, Wiley. (**c**) Schematic illustration of capillary microfluidics chip for precisely controlled generation of monodisperse double and triple emulsions and optical images of monodisperse double and triple emulsions containing controlled numbers of droplets. Reproduced with permission [85]. Copyright 2007, Wiley.

### 3.3. Lipid Nanoparticles for Drug Delivery

Benefitting from the high biocompatibility of lipid molecules, liposome and lipid nanoparticles (LNPs) are widely used as drug delivery systems. Liposomes are composed of lipid bilayers with a hollow structure, enabling them to encapsulate a variety of drugs including hydrophilic, hydrophobic, and lipophilic drugs. The fabrication methods of liposomes and LNPs can be divided into hydration method, organic solvent injection method, nanoprecipitation method and reversed-phase evaporation vesicle method [86]. Among them, the nanoprecipitation method is to dissolve lipids in organic solvents and inject them into a high-speed stirring aqueous phase to form liposome or LNPs. In general, lipids are dissolved in water-miscible organic solvents such as methanol and ethanol. Then, the two phases are mixed rapidly. In this case, the rapidly changing polarity supersaturates lipids, causing the lipids to self-assemble to form liposomes or LNPs (Figure 5a) [87]. 

A microfluidic mixer is a microfluidic chip that can efficiently control fluid-mixing by designing a microstructure. For flat microchannel, viscous forces dominate over inertial forces due to the low Reynolds number. The liquids exhibit laminar and uniaxial flow. Laminar liquids are mixed through molecular diffusion. This process is time-consuming, and has poor uniformity, which reduces the mixing efficiency [88]. To overcome this drawback, Zhigaltsev et al. used a staggered herringbone mixer (SHM) to fabricate size-constrained LNPs (Figure 5b) [89]. The asymmetrically arranged herringbone groove patterns cause the passing fluid to rotate and stretch to generate chaotic flow (Figure 5c) [90]. For a predetermined microfluidic mixer, the total flow rate and flow ratio can significantly affect the size and uniformity of LNPs. Specifically, the increase in total flow rate and flow ratio resulted in smaller particle size and higher uniformity [91].

Nucleic acids such as small interfering RNA (siRNA) and messenger RNA (mRNA), are negatively charged and easily degraded by ubiquitous RNases. The delivery of nucleic acids requires carriers to protect them from degradation and facilitate cellular uptake. However, traditional liposomes cannot achieve lysosomal escape. Cationic liposomes and LNPs have been used to achieve lysosomal escape. In microfluidic mixer chips, lipids in ethanol are rapidly mixed with siRNA in low pH (pH = 4) buffer media. LNPs are composed of ionizable lipids, helper lipids, cholesterol, and polyethylene glycol lipids. During mixing process, ionizable lipids are positively charged under acidic conditions and bind to nucleic acids through electrostatic interaction [92]. Unlike liposomes, LNPs-siRNA exhibited electron-dense core structures (Figure 5d) [93]. Benefiting from electrostatic interaction, the encapsulation efficiency of RNA was always more than 90%. In recent years, mRNA vaccines have developed rapidly. In particular, two mRNA vaccines, mRNA-1273 and BNT162b2 developed by Moderna and BioNTech/Pfizer, respectively, have received emergency authorizations for widespread clinical use against the SARS-CoV-2 virus [94,95,96]. Compared with traditional vaccines, mRNA vaccines have no potential risk of infection or genome integration. Uri et al. encapsulated SARS-CoV-2 human Fc-coupled receptor binding domain (RBD-hFc) mRNA into LNPs (Figure 5e) [97]. After intramuscular administration, the LNPs-mRNA vaccine triggered strong humoral responses, high levels of neutralizing antibodies, and Th1-biased cellular response in BALB/c mice.

**Figure 5 bioengineering-09-00625-f005:**
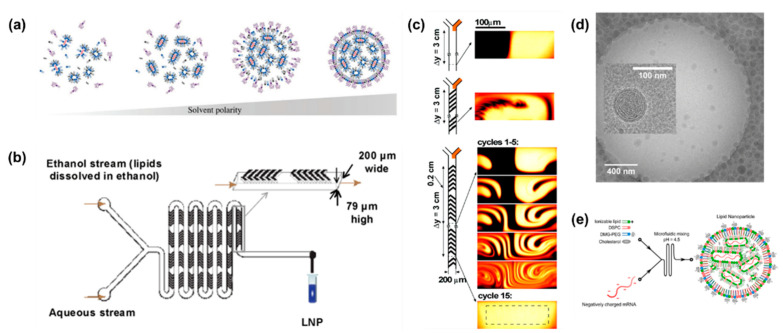
(**a**) Schematic diagram of mechanism for LNPs preparation by nanoprecipitation. When lipid-laden organic solvents and RNA-laden aqueous solutions are intermixed, ionizable lipids and negatively charged nucleic acids bind to each other via electrostatic interactions to form the core, and other lipids cover the surface to form lipid monolayers. Reproduced with permission [87]. Copyright 2018, Wiley. (**b**) Schematic illustration of LNPs preparation by SHM. Lipid-laden ethanol is introduced through one inlet of the SHM, whereas aqueous solution is introduced through the other inlet. Reproduced with permission [89]. Copyright 2012, American Chemical Society. (**c**) Schematic illustration of channel mixing behaviors for wall-less structures, straight ridges, and interlaced herringbone structures. Reproduced with permission [90]. Copyright 2002, American Association for the Advancement of Science. (**d**) Cryo-transmission electron microscopy images of LNPs-siRNA, showing an electron-dense core structure. Reproduced with permission [93]. Copyright 2012, American Chemical Society. (**e**) Schematic illustration of RBD-hFc mRNA LNP synthesis. Reproduced with permission [97]. Copyright 2020, American Chemical Society.

SHM has been commercialized by Precision Nanosystems for many years. Scientific research based on this commercial chip has been widely reported. Alternative types of microfluidic mixers including microfluidic fluid dynamic flow focusing, microfluidic Tesla mixer, microfluidic bifurcating mixers and baffle mixers have also been employed for LNPs preparation [98]. In addition to LNPs, polymers soluble in water-miscible organic solvents can also be used to prepare nanoparticles by this method.

### 3.4. Other Micro/Nanocarriers for Drug Delivery

In addition to micro/nanogels, microcapsules, and LNPs, many exquisitely designed hierarchical micro/nanoparticles have also been used for drug delivery, including porous, Janus, and multi-compartment micro/nanoparticles. 

Compared with the solid ball, mesoporous spheres have a larger surface area, making it easier for them to release drugs from inside. Due to their high specific surface area and high loading capacity, porous materials are ideal candidates for drug carriers. Gu et al. fabricated spherical mesoporous colloidal photonic crystal particles (MCPCP) by self-assembly of monodisperse mesoporous silica nanoparticles (MSN) within microfluidic droplet templates (Figure 6a) [99]. The synthesized particles were tightly packed by MSN to form a highly ordered arrangement. In this case, the nanopores of MSN facilitated the loading of small molecules such as DOX, and the highly ordered macroporous structure enables the loading of macromolecular drugs such as albumin-bound paclitaxel. Furthermore, thermosensitive PNIPAM hydrogel can be perfused into the MCPCP voids to regulate drug release and prevent drug loss during transport.

Zhao et al. prepared hierarchical porous microparticles composed of PLGA and drug-loaded hollow porous MSN [100]. During PLGA curing, internal droplets leaked out of the double emulsion template to form microparticles with single or multiple opened structures (Figure 6b). The drug release rate was adjusted by the number of surface openings. Microparticles loaded with desferrioxamine and porcine decellularized dermal matrix formed a composite scaffold to treat partial abdominal wall defects of rat. Eight weeks after implantation, the release of desferrioxamine from the scaffold significantly accelerated neointima formation and intense collagen deposition.

Wang et al. fabricated a polymer/inorganic nanoparticle hybridized Janus-like vesicle motor without template [101]. This Janus-like vesicles motor was divided into two distinct halves, amphiphilic poly(ethylene oxide)-b-poly(styrene) (PEO-b-PS) and inorganic nanoparticles such as Pt or Au nanoparticles tethered to sulfhydryl groups (Figure 6c). Hybrid vesicles containing asymmetrically distributed Pt nanoparticles can catalyze decomposition of hydrogen peroxide (H_2_O_2_), thereby propelling themselves into relatively oriented motion. Due to the excellent photothermal response of gold nanoparticles, hybrid vehicles were damaged by localized high temperature under NIR irradiation (808 nm). Inorganic nanoparticles were selectively incorporated into vesicles to confer versatility in vesicles.

In addition, due to the differences in the composition of Janus micro/nanoparticles, Janus micro/nanoparticles have great potential for applications in dual-stimulus responsive drug carriers. Kim et al. synthesized dual stimuli-responsive drug carriers for pH-sensitive and redox-sensitive compartments using stop-flow lithography [102]. The model drugs were released independently upon exposure to the corresponding stimuli (Figure 6d). In brief, under redox conditions, the redox-responsive compartment was completely degraded, while the other compartment remained intact. However, compartmental degradation was reversed when Janus particles were incubated under the acidic conditions. Both model drugs were effectively released under pH and redox stimulation. Micro/nanodelivery systems were summarized as in Table 1.

**Figure 6 bioengineering-09-00625-f006:**
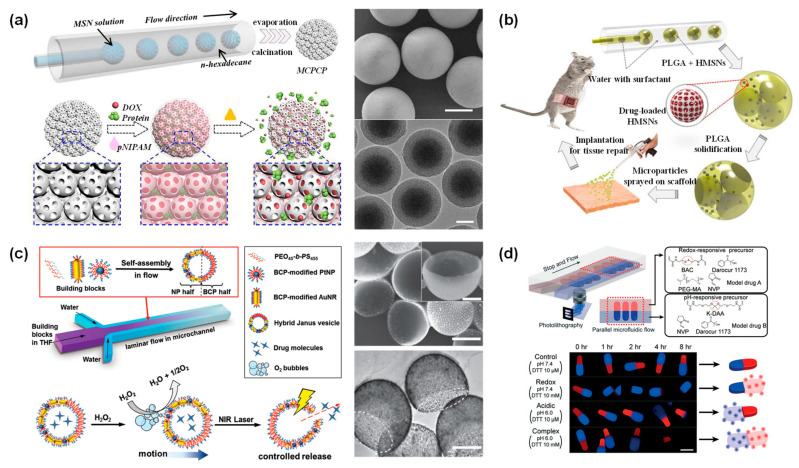
(**a**) Schematic illustration of MCPCP production by microfluidic chip. SEM images of MCPCP and TEM image of the MSN. Reproduced with permission [99]. Copyright 2018, American Chemical Society. (**b**) Schematic diagram of capillary microfluidic chip for fabrication of hierarchically porous composite microparticles and application in abdominal wall repair. Reproduced with permission [100]. Copyright 2018, Royal Society of Chemistry. (**c**) Schematic illustration of self-assembly from a mixture of PEO-b-PS and PEO-b-PS modified Au/Pt nanoparticles into Janus-like vesicles. The self-propelled scheme in the presence of H_2_O_2_ and the controlled release scheme under NIR light irradiation. SEM and TEM images of vesicles. Reproduced with permission [101]. Copyright 2015, Wiley. (**d**) Dual stimulus response Janus drug carrier precursors aligned in parallel microchannel; fluorescence images of microparticles under acidic or redox stimulation. Reproduced with permission [102]. Copyright 2018, Royal Society of Chemistry.

### 3.5. Lymphatic Chip for Drug Delivery

The development of micro/nanoparticle delivery systems has achieved some clinical success, such as Doxil, mRNA-1273 and BNT162b2. However, the current understanding of the behavior of delivery carriers in complex physiological environments is limited, which restricts the optimal design of delivery carriers. In addition, 2D cell culture system cannot simulate complex interactions in physiological environments [143]. Therefore, the development of 3D cell culture models has attracted more and more attention. The diameter of microfluidic channels is suited for the physical dimensions of different kinds of cells. In addition to manufacturing multifunctional functional delivery carriers, microfluidic chips can also construct in vitro microphysiological systems to simulate organ-level and even organism-level functions. 

Various microfluidic cell culture chips, also known as organs-on-a-chip, are designed for precision medicine [144,145,146]. However, most organs-on-a-chip often ignore the role of lymphatic drainage. The lymphatic system plays an important role in circulatory and immune systems [147]. Lymphatic vessels are composed of initial lymphatic vessels and collecting lymphatic vessels. The fundamental function of lymphatic vessels is to maintain the balance of interstitial fluid. Due to the limited transport capacity of blood vessels, interstitial fluid cannot be completely reabsorbed through the blood vessels and returned to the vascular system. The static interstitial fluid brings a pressure gradient, causing the fluid flow into the initial lymphatic vessels. However, excellent transport capacity may cause unexpected implications. For example, tumor cells are more likely to metastasize through the lymphatic system. Invasive tumor cells tend to rely more on highly permeable lymphatic systems for metastasis. Given the important role of lymphatic vessels in maintaining tissue fluids homeostasis, simultaneous construction of vascular and lymphatic systems is important for modeling cancer pathology. Tatsuya et al. used a perfusion culture method to co-culture human umbilical vein endothelial cells and lymphatic endothelial cells (LECs) in collagen gel embedded microfluidic platform with 600 μm diameter parallel channels (Figure 7a) [148]. Lymphatic vessels have significant differences in intercellular adhesion with blood vessels. The lymphatic capillaries exhibited a discontinuous button-like structure, while the blood vessels and the collecting lymphatic vessels showed a continuous zipper-like structure. Interspaces at the junctions of the button-like structure act as a valve to regulate the passage of immune cells, proteins and interstitial fluid. The in vitro reconstruction of lymphatic vessels structures helps to better simulate organ physiology and disease status. On a more complex microfluidic chip, blood vessels were also integrated into the tumor-lymphatic vessels model [149]. Cao et al. fabricated hollow blood vessels with an opening on both ends and lymphatic vessels with blind holes at one end by bioprinting technology (Figure 7b). The two-vessel chip has faster drug transport than a single vessel due to the accelerated drug transport in lymphatic vessels. Therefore, under DOX perfusion, the survival rate of MCF-7 cells in both vascular systems can be maintained at more than 90%. The high cell vitality was because of the timely transport of DOX by lymphatic vessels, which reduced the accumulation of DOX. This biomimetic design can simulate transport kinetics of drugs in biological tissues to some extent. 

Vaccines are typically administered subcutaneously or intramuscularly, where local antigen-presenting cells (usually dendritic cells) uptake and process antigens for subsequent delivery to T cells in secondary lymphoid organs (usually lymph nodes). During the transport process, lymphatic vessels are required. Alexandra et al. investigated the 3D chemotaxis of bone marrow-derived dendritic cells (BMDCs) mediated by different Toll-like receptor adjuvants, such as MPLA (TLR4 ligand) and CpG (TLR9 ligand), in a 3D collagen matrix-based microfluidic device (Figure 7c) [150]. Free MPLA increased the chemotaxis of BMDC to chemokines CCL19 and CCL21 gradients, while free CpG was ineffective. However, CpG-encapsulated PLGA microparticles enhanced BMDC chemotaxis. Dendritic cells secreting CCL19 and LECs secreting CCL21 play an important role in the migration of dendritic cells from peripheral tissues to lymph nodes. Since vaccines require dendritic cells to present antigens to T cells in lymph nodes, modulating DC migration to lymph nodes with different adjuvants or combinations will facilitate vaccine development. Currently, the 2D cell culture models are too simple and mouse models are too complex. There is a lack of models to study the response behavior between drugs and intact tissue substructures. Ashley et al. investigated the specific localization of molecular stimuli generated by drug delivery in the B-cell or T-cell zone of lymph nodes (Figure 7d) [151]. Fluorescent dextran with two molecular weights (40 kDa and 70 kDa) reaching specific substructures of lymph nodes were studied. The T-cell zone of the lymph node retained the model drug and glucose-coupled albumin larger than the B-cell zone. The response of different substructures in the lymph nodes to external signals is variable. Investigating the differences in response of lymph node substructure regions to vaccine delivery can promote understanding of local immune responses. Lymphatic vessel models were summarized as in Table 2.

**Figure 7 bioengineering-09-00625-f007:**
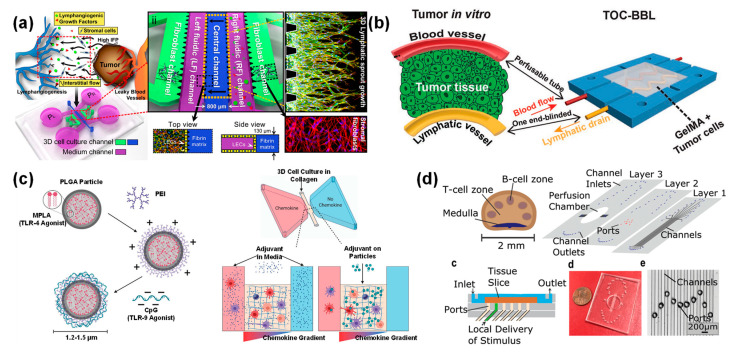
(**a**) Schematic diagram of microfluidic chip for interstitial flow-induced lymphangiogenesis. The central channel was filled with fibrin matrix for loading LECs; the outermost channels were loaded with fibroblasts to mimic paracrine effects. Reproduced with permission [148]. Copyright 2016, Elsevier. (**b**) In vitro model with both opened vessels and blinded lymphatic vessels at one end on a tumor microfluidic chip. Reproduced with permission [149]. Copyright 2019, Wiley. (**c**) MPLA was encapsulated inside the PLGA microparticles and CpG was bound to the microparticle surface by electrostatic interaction. Schematic diagram of the microfluidic device designed to study BMDC migration behavior. BMDC are loaded in the central collagen matrix channel, while chemokines are loaded in the left chamber and diffuse towards the middle channel. Reproduced with permission [150]. Copyright 2021, Wiley. (**d**) Microfluidic device for studying the effect of local stimulation on lymph nodes. Schematic of the exploded view and side view of the device and photographs and micrographs of the assembled device. Reproduced with permission [151]. Copyright 2017, Royal Society of Chemistry.

## 4. The Challenges of Microfluidic Technology 

Benefiting from the rapid development of micro/nano fabrication technology, micro/nanoparticles with high dispersibility and reproducibility can be fabricated by microfluidic devices. Microfluidics faces several unavoidable challenges in the field of micro/nanoparticles synthesis. Manufacturing devices with complex geometries and configurations increases manufacturing costs and processing complexity. Despite the low cost and ease-of-preparation characteristics of capillary microfluidics, large batch-to-batch variability limits its controllability. One solution is parallelized production, which can improve production efficiency by paralleling multiple microfluidic chips. Another solution is to increase the throughput of generated micro/nanoparticles. It is also necessary to consider that the pressure in the microfluidic channel increases rapidly with the increase of the flow rate, which poses a challenge to the packaging technology of microfluidic device. In conclusion, microfluidics enables fabrication of micro/nanoparticles with specific structures and functions for drug delivery. 

On the other hand, the development of the organ-on-a-chip provides an in vitro drug-screening model. The role of the lymphatic system, as mentioned previously, tends to be overlooked. In vitro construction of tissues, blood vessels and lymphatic vessels increases the complexity of the organ-on-a-chip, resulting in very low throughput. However, drug delivery behavior requires high-throughput systems to verify statistically significant studies. Microfluidic chip materials, such as PDMS, may adsorb hydrophobic small molecule drugs, making them unable to be effectively used in drug research. One of the effective methods to solve this problem is to reduce adsorption by surface modification.

## 5. Future Perspectives

Benefiting from the sophisticated control of microscale fluids, microfluidics allows the modulation of the physicochemical properties (e.g., composition, shape and surface ligands) of the micro- and nano-delivery delivery vehicles to be tuned during fabrication. Well-designed delivery vehicles can optimize drug delivery efficiency and release profiles. These well-thought-out designs help reduce adverse effects and improve outcomes during treatment.

While combating the global spread of SARS-CoV-2, the mRNA-LNP vaccine was rapidly approved and available for clinical treatment. In addition, the mRNA-LNP vaccines were fabricated on a large scale by microfluidic mixer. This has greatly encouraged people to use microfluidic technology to prepare drug delivery vehicles on a large scale. It provides valuable guidance for the preparation of micro/nano delivery vehicles using microfluidic mixing technology. In addition, this technique can be extended for the preparation of polymer nanoparticles. At present, the preparation of microparticles mainly focuses on droplet microfluidics. As mentioned previously, droplet microfluidics limits microparticles preparation throughput and batch-to-batch variability. Droplet microfluidics often requires the use of organic solutions, which will certainly limit the clinical treatments. Incomplete removal of organic solvents may be detrimental to patient health. The presence of organic solutions may also limit the delivery of bioactive drugs. For example, during encapsulation, enzyme denaturation was observed at the oil–water interface. Therefore, the large-scale production of microparticle delivery vehicles in a simple, controllable and highly biocompatible manner is still required in the future. Thermodynamically equilibrium immiscible aqueous phase solutions can be used in place of organic phase solutions. There have been some studies using all-aqueous emulsions as templates to prepare microparticles.

In vitro construction of disease models can help reduce the time it takes for drug discovery. Currently developed organs-on-chips often lack the vascular system, especially lymphatic vessels, which are often overlooked in the construction of disease models. Oversimplified models cannot simulate the physiological environment of lymphatic vessels in vivo. Considering the significant differences between animal models and human physiological states, there is still great hope in constructing rational disease models on microfluidic chips to replace animal models for drug screening. Most reported lymphatic vessel models focus on the study of lymphangiogenesis and lymphatic permeability in primary lymphatic vessels. In the future, it may be possible to introduce smooth muscle-mediated lymphatic pumping mechanisms to collect lymphatic vessels to study the behavior of active drug delivery.

## 6. Conclusions

This review first discusses the advantages and disadvantages of typical microfluidic chip materials and processing techniques. Subsequently, micro/nanoparticle drug delivery carriers, including micro/nanogels, microcapsules, LNPs, and hierarchical micro/nanoparticles, were prepared using a variety of microfluidic chips. The microfluidic chip structure design and material selection can endow the delivery vehicle with a variety of stimuli responsiveness, such as acid, pH, temperature, light, osmotic pressure, etc. Micro/nano drug delivery systems offer highly tunable drug release patterns, allowing them to incorporate multiple therapeutic modalities. In addition, we summarized the lymphatic–drug interactions that are easily overlooked in organ-on-a-chip construction. Therefore, well-designed microfluidic systems have great potential for designing effective drug development and personalized medicine.

## Figures and Tables

**Table 1 bioengineering-09-00625-t001:** Stimuli-responsive strategies for micro/nanoparticles drug delivery systems.

Drug Delivery Systems	Materials	Stimuli	Ref.
Micro/nanogels	sodium alginate; poly(vinyl alcohol);	pH	[57,103]
	hyaluronic acid; Methacrylate-modified hyaluronic acid	hyaluronidase	[60,67]
	PNIPAM and Pyy; poly(N-isopropylacrylamide-coacrylic acid); PEGylated poly(2-(N-morpholino) ethyl methacrylate)	temperature	[66,104,105]
	PEO-based triblock copolymers	structure and ionic nature	[106]
	poly (ethylene glycol); N-(2-oxopropyl)methacrylamide (OPMA)	sustained release	[107,108]
	poly(ferrocenylsilane)	redox	[109]
Microcapsules	decanol, lauryl acrylate and trimethylolpropane ethoxylate triacrylate (ETPTA); GMA, ETPTA and 1-decanol	polarity	[75,110]
	PDMS, silicon oil and curing agent;	electric fields	[77]
	poly(ethylene glycol) divinyl ether and trimethylolpropane tris(3-mercaptopropionate); PEG and dextran; perfluoropolyether; PDMS, silicon oil and curing agent;	osmolarity	[81,111,112,113]
	poly(ethylene glycol) diacrylate (PEGDA); ethoxylated trimethylolpropane triacry-late	multifunctional responsive	[76,114]
	chitosan and PLGA; PLGA; alginate sodium, resistant starch and chitosan; terephthalaldehyde-crosslinked chitosan; shellac	pH	[82,115,116,117,118]
	PEGDA and PNIPAM; sodium alginate and PEG-g-chitosan; palm oil; cellulose acetate	temperature	[83,119,120,121,122]
	chitosan, terephthalaldehyde (TPA) and magnetic NPs; iron oxide NPs and chi-tosan; polyethylene glycol, polyĲdial-lyldimethylammonium chloride) and ferrofluid	magnetic	[111,123,124]
	perfluorocarbon and sodium alginate; PEGDA	ultrasound	[125,126,127]
	oppositely charged polyelectrolyte; PLGA and PCL photocurable resin	sustained release	[128,129,130]
Other micro/nanocarriers	dextran-rich core and a tetra-PEG; PLGA and PCL; PLGA and MSNs; porous silicon (PSi) microparticles and lipid; poly(acrylamide) and poly(methyl acrylate)	sustained release	[100,131,132,133,134]
	calcium alginate; poly(methyl vinyl ether-co-maleic acid) and porous silicon NPs; ketal-containing diacrylamide	pH	[135,136,137]
	MCPCP and PNIPAM; PNIPAM and PNIPMAM; poly(N-vinyl caprolactam)	temperature	[99,138,139]
	PEO-b-PS and BCP modified AuNRs/PtNPs; PVCL and Au NPs; azopyridyl polymer	light	[101,140,141]
	ketal linkage-containing precursor and a reducible disulfide linkage-containing precursor; poly(N-isopropylacrylamide-co-3-aminophenylboronic acid-co-acrylic acid)	multifunctional responsive	[102,142]

**Table 2 bioengineering-09-00625-t002:** In vitro reconstitution of lymphatic vessel models.

Type	Cells	Major Research	Ref.
Lymphangiogenesis models	LECs; LECs and HUVECs	interstitial flow	[148,152]
	LECs and HUVECs	bi-directional effects between the lymphatic and vascular cells	[153]
	LECs	lymphatic barrier function	[154]
	LECs and blood vascular endothelial cells	permeability	[155]
Lymphatic-tumor models	MCF7 cells, MDA-MB-231 cells and LECs	pathological changes	[156]
	MCF-7 cells	diffusion profiles for biomolecules and drugs	[149]
	LECs and MDA-MB-231 cells	extracellular matrix density	[157]
	Tumor-derived fibroblasts and tubular lymphatic vessel	tumor progression and lymph node metastasis	[158]
	LECs and MDA-MB-231	flow enhanced invasion	[159]
	tumor-draining lymph nodes and tumor	tumor-lymph node interaction	[160]
Lymphatic-organ models	DCs	chemotaxis	[150]
	lymph node slices	local stimulation	[151]

## Data Availability

Data sharing is not applicable to this review.

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
