# Peer review of "Recent Advances in Drug Delivery System Fabricated by Microfluidics for Disease Therapy"

_bioengineering, 2022, doi:10.3390/bioengineering9110625_

Round 1

Reviewer 1 Report

This is a well-written concise, yet comprehensive review on the utility of microfluidic technology for drug delivery applications. The authors discuss the microfluidic technology itself, and describe relevant examples of a few different types of drug delivery particles. The article will be significantly strengthened and help a wider range of readers if the following two points are addressed:

1. In many of the drug delivery examples discussed, it rather reads like this is a review on drug delivery micro/nanoparticles, and not necessarily from microfluidic technology point of view. It would be very helpful if there are more discussions for each example on how and more importantly why microfluidic technology is used to fabricate the specific particles. What were the disadvantages of similar particles made with conventional, non-microfluidic strategies, and how did microfluidics overcome the issue?

2. There are new concepts introduced in the conclusion section, namely the challenges of microfluidic technology. It would be better to have a separate, dedicated section for this discussion, and allow conclusion to summarize the article.

Author Response

Point-by-point response to reviewers

We would like to thank all the reviewers for their insightful and thoughtful comments! We have revised the manuscript according to their advices, which should significantly improve the clarity and quality of our work. Below is a list of the point-by-point responses to the reviewer comments and the corresponding changes that we made. All the changes are highlighted in the manuscript.

Reviewer #1: This is a well-written concise, yet comprehensive review on the utility of microfluidic technology for drug delivery applications. The authors discuss the microfluidic technology itself, and describe relevant examples of a few different types of drug delivery particles. The article will be significantly strengthened and help a wider range of readers if the following two points are addressed:

Re: We thank the reviewer for the thoughtful comments!

  1. In many of the drug delivery examples discussed, it rather reads like this is a review on drug delivery micro/nanoparticles, and not necessarily from microfluidic technology point of view. It would be very helpful if there are more discussions for each example on how and more importantly why microfluidic technology is used to fabricate the specific particles. What were the disadvantages of similar particles made with conventional, non-microfluidic strategies, and how did microfluidics overcome the issue?

Re: As per the advice, we have added the discussions of disadvantages of nanoparticles prepared by conventional, non-microfluidic strategies in section 3.1 and 3.2, and discussed the advantages of using microfluidic devices. Because LNPs are normally prepared using microfluidic devices, we did not add discussion in section 3.3. Accordingly, the advantages of using microfluidic devices to prepare nanoparticles to overcome issues of traditional synthesis methods were also discussed in Introduction section. The aforementioned discussion is now incorporated in lines 57-68 on page 2, lines 263-272 on page 6, lines 373-381 on pages 8-9 of this revision.

  1. There are new concepts introduced in the conclusion section, namely the challenges of microfluidic technology. It would be better to have a separate, dedicated section for this discussion, and allow conclusion to summarize the article.

Re: Per the advice, we have included a separate section in this revision discussing the challenges of microfluidics and have rewritten the conclusions. The aforementioned discussion is now incorporated in lines 526-535 on pages 12-13 of this revision. Again, we thank the reviewer for the insightful and thoughtful comments.

Reviewer 2 Report

Comments on Bioengineering-1920761:

This review presents the: Recent Advances in Drug Delivery System Fabricated by Mi- 2crofluidics for Disease Therapy.

A quick search in the Scopus/web of science using microfluidic and drug delivery keywords shows that there are numerous review articles regarding microfluidic drug delivery and therefore the current review does not add anything more to the available review papers. The outlines covered in the current manuscript such as micro/nano carriers, microcapsules, lipid NPs, gels, etc, are already covered in the previously published review papers. Therefore, this manuscript is not suitable for publication in the current form.

Author Response

Point-by-point response to reviewers

We would like to thank all the reviewers for their insightful and thoughtful comments! We have revised the manuscript according to their advices, which should significantly improve the clarity and quality of our work. Below is a list of the point-by-point responses to the reviewer comments and the corresponding changes that we made. All the changes are highlighted in the manuscript.

Reviewer #2: This review presents the: Recent Advances in Drug Delivery System Fabricated by Microfluidics for Disease Therapy.

A quick search in the Scopus/web of science using microfluidic and drug delivery keywords shows that there are numerous review articles regarding microfluidic drug delivery and therefore the current review does not add anything more to the available review papers. The outlines covered in the current manuscript such as micro/nano carriers, microcapsules, lipid NPs, gels, etc, are already covered in the previously published review papers. Therefore, this manuscript is not suitable for publication in the current form.

Re: We thank the reviewer for the critical comments! Yes, there have been many review articles on microfluidic drug delivery discussed, which shows that the rapid development of microfluidic technology in recent years has drawn extensive attention to the field of drug delivery. In addition to some micro/nanocarriers, microcapsules and LNPs prepared by microfluidic devices, some useful information from this review is detailed below.

  1. In this review, we analyzed the preparation of drug delivery systems from microparticles to nanoparticles and hierarchical micro/nanoparticles and their applications in materials science.
  2. We emphasize the preparation of multifunctional delivery systems by selecting appropriate materials and processing techniques, rather than repeatedly demonstrating delivery systems with the same functionality. This versatile delivery system could significantly reshape drug delivery patterns to aid the development of precision medicine.
  3. We highlight the impact of microstructure, nanostructure and micro/nanostructure design perspectives on drug delivery behavior.

With these clarifications, we sincerely hope that the reviewer agrees that the novelty of this review is high enough and appropriate for publication in Bioengineering.

Reviewer 3 Report

The review on micro-/nanofluidic drug delivery is a sound overview on manufacturing such systems and gives examples for their applicability. Maybe, to put it more into perspective, it would be good to clarify, where the frontline to traditional, macroscopic delivery systems is and how these nanomaterials are generally applied opposed to the traditional devices/ways for drug delivery. Also, give some examples for release types that do not come naturally such as electrical field changes. It was not clear to me, how this can be applied in vivo. Other than that I have no complaints.

Author Response

Point-by-point response to reviewers

We would like to thank all the reviewers for their insightful and thoughtful comments! We have revised the manuscript according to their advices, which should significantly improve the clarity and quality of our work. Below is a list of the point-by-point responses to the reviewer comments and the corresponding changes that we made. All the changes are highlighted in the manuscript.

Reviewer #3:  The review on micro-/nanofluidic drug delivery is a sound overview on manufacturing such systems and gives examples for their applicability. Maybe, to put it more into perspective, it would be good to clarify.

Re: We thank the reviewer for the thoughtful comments!

  1. Where the frontline to traditional, macroscopic delivery systems is and how these nanomaterials are generally applied opposed to the traditional devices/ways for drug delivery.

Re: As per the advice, we have added a comparison of macroscopic and micro/nanoparticle drug delivery systems to the Introduction. Currently, both small drugs and macromolecular drugs require the appropriate drug delivery strategies [6]. For example, traditional small molecule drugs tend to have low solubility, which limits their bioavailability. Furthermore, the in vivo stability of protein drugs after administration is compromised by proteases, temperature and pH. Nucleic acid drugs need to be delivered into the cytoplasm to be effective. To address these chal-lenges, it is necessary to develop appropriate drug delivery systems. Proper drug delivery allows not only on-demand delivery of active drug to target tissues or cells, but also proper control of pharmacokinetics (eg, in vivo distribution, half-life, maximum concentration in serum, etc.) [7]. Although pharmacokinetics can be modulated by controlling the number and frequency of dosing, many disease treatments require frequent long-term dosing, which compromises patients' quality of life. A variety of controlled release systems in-cluding injectable hydrogels, polymer implants and micro/nanoparticles have been used to improve drugs delivery behaviors [8]. Drugs within a delivery system have multiple re-lease modes, including drug diffusion, degradation of the delivery material, and external stimuli. Macroscopic delivery systems such as drug-eluting stents require surgical im-plantation and may induce fibrocystic reaction [9]. And the macroscopic delivery system has a singular drug release pattern. Compared with macroscopic delivery systems, micro/nanoparticle delivery systems offer many advantages in protecting protein and nucle-ic acid drugs from degradation, controlling drug release profiles, enhancing small mole-cule solubility of hydrophobic drugs, and tunable tissue targeting [10]. The aforementioned discussion is now incorporated in lines 37-56 on page 1-2 of this revision.

  1. Also, give some examples for release types that do not come naturally such as electrical field changes. It was not clear to me, how this can be applied in vivo. Other than that, I have no complaints.

Re: Sorry for the confusion. Electric fields can also be used to control the release of active substances in microcapsules by methods including electroporation and iontophoresis. Depending on the location of the device, drug release near the skin can be triggered by a voltage generat-ed by an external conductive skin patch. Alternatively, a minimally invasive implantable device controlled by radio frequency can be used to control drug release. The aforementioned discussion is now incorporated in lines 297-302 on page 7 of this revision. Again, we thank the reviewer for the insightful and thoughtful comments.

Round 2

Reviewer 1 Report

The authors have addressed my comments adequately. The manuscript is in good form, and would attract a wide range of audience. I have no further comments.

Author Response

Please find the response in the attached file.

Reviewer 2 Report

I have read the revised manuscript and the authors responses to my comments and still believe this manuscript does not add anything new to the previously published review papers in the present form.

Author Response

(The authors gave the same response as above.)
